# Alzheimer’s Disease Associated *Presenilin 1* and *2* Genes Dysregulation in Neonatal Lymphocytes Following Perinatal Asphyxia

**DOI:** 10.3390/ijms22105140

**Published:** 2021-05-13

**Authors:** Agata Tarkowska, Wanda Furmaga-Jabłońska, Jacek Bogucki, Janusz Kocki, Ryszard Pluta

**Affiliations:** 1Department of Neonate and Infant Pathology, Medical University of Lublin, 20-093 Lublin, Poland; agatatarkowska@umlub.pl (A.T.); wm.jablonska@gmail.com (W.F.-J.); 2Department of Organic Chemistry, Medical University of Lublin, 20-093 Lublin, Poland; jacekbogucki@wp.pl; 3Department of Clinical Genetics, Medical University of Lublin, 20-080 Lublin, Poland; januszkocki@umlub.pl; 4Laboratory of Ischemic and Neurodegenerative Brain Research, Mossakowski Medical Research Institute, Polish Academy of Sciences, 02-106 Warsaw, Poland

**Keywords:** perinatal asphyxia, hypoxic-ischemic encephalopathy, brain ischemia, Alzheimer’s disease, lymphocytes, *amyloid protein precursor*, *β-secretase*, *presenilin 1* and *2*, hypoxia-inducible factor 1-α, genes

## Abstract

Perinatal asphyxia is mainly a brain disease leading to the development of neurodegeneration, in which a number of peripheral lesions have been identified; however, little is known about the expression of key genes involved in amyloid production by peripheral cells, such as lymphocytes, during the development of hypoxic-ischemic encephalopathy. We analyzed the gene expression of the *amyloid protein precursor*, *β-secretase*, *presenilin 1* and *2* and *hypoxia-inducible factor 1-α* by RT-PCR in the lymphocytes of post-asphyxia and control neonates. In all examined periods after asphyxia, decreased expression of the genes of the *amyloid protein precursor*, *β-secretase* and *hypoxia-inducible factor 1-α* was noted in lymphocytes. Conversely, expression of *presenilin 1* and *2* genes decreased on days 1–7 and 8–14 but increased after survival for more than 15 days. We believe that the expression of *presenilin* genes in lymphocytes could be a potential biomarker to determine the severity of the post-asphyxia neurodegeneration or to identify the underlying factors for brain neurodegeneration and get information about the time they occurred. This appears to be the first worldwide data on the role of the *presenilin 1* and *2* genes associated with Alzheimer’s disease in the dysregulation of neonatal lymphocytes after perinatal asphyxia.

## 1. Introduction

The prevalence of perinatal asphyxia is ~2 per 1000 live births in developed countries, but the number of those affected rises to 26 per 1000 in developing countries [1,2,3,4,5,6]. Perinatal asphyxia occurs worldwide in approximately 4 million newborns annually [7] and is responsible for 23% of all infant deaths and 8% of child deaths [8]. Up to 25% of survivors have persistent neurological deficits [2], which is the second most common cause of neurological disability in the world [3]. Despite advances in treating the causes of perinatal asphyxia and asphyxia alone, the number of perinatal asphyxia cases has not decreased over the last decade [9], and its consequence is the development of hypoxic-ischemic encephalopathy diagnosed 1.8 per 1000 live births [10,11,12] with further deterioration the postpartum neurological status of newborns. Hypoxic-ischemic encephalopathy can result in severe neurodevelopmental disability and mortality in, respectively, 24.9% and 34.1% of cases [13]. Further observations of infants suffering from perinatal asphyxia showed that 27–33% of them aged 6–7 years will have features of intellectual disability [14]. This brain damage is secondary to an episode of hypoxia-ischemia as well as reoxygenation-reperfusion following resuscitation [15]. Hypoxic-ischemic encephalopathy development post-asphyxia in newborns is still a common and serious disease entity. It is well known that infants with hypoxic-ischemic encephalopathy carry a high risk of adverse effects later in life, such as neurodegenerative disorders, cognitive changes, severe mental retardation, learning and behavior problems and motor impairments, which develop as long-term consequences [1,9]. How to prevent brain injury from hypoxic-ischemic encephalopathy remains an important issue [1,4]. Hypoxic-ischemic encephalopathy involves a combination of reduced oxygen supply to the blood (i.e., hypoxia) and reduced blood flow in the brain (i.e., ischemia) [1,4]. Above lead to the immediate death of neurons (necrosis) during asphyxia and delayed death of neuronal cells (apoptosis) which can occur after an injury from asphyxia [12] and finally these processes significantly contribute to the permanent brain injury and atrophy mainly in postpartum period [15,16,17,18].

Perinatal asphyxia in mice causes delayed damage to the hippocampus and a significant deficit in spatial learning and memory [19]. Significantly higher levels of amyloid protein precursor, increased total tau protein and tau protein phosphorylation, decreased hypoxia-inducible factor, lower levels of amyloid-degrading neprilysin, increased amyloid deposition with activation of astrocytes and microglia in the brain have also been documented [20,21]. To the best of our knowledge, one study has assessed that blood tau protein levels can be used as a predictor of the neurological prognosis of human neonatal asphyxia [22]. These data suggest that the hallmarks of Alzheimer’s disease may begin during pre- or perinatal asphyxia and progress to full-blown disease later in life, so these conditions may contribute to the development of Alzheimer’s disease, supporting the notion that environmental factors may cause or worsen symptoms of Alzheimer’s disease [20,21,23]. It has been suggested that cerebral ischemia and neonatal hypoxia-ischemia may be causally related to Alzheimer’s disease [24,25,26,27,28,29]. Brain ischemia induces neurodegenerative alterations associated with the development of cognitive impairments, such as in Alzheimer’s disease [24,25,26,27,28,29,30]. The mechanisms by which Alzheimer’s disease develops after brain ischemia and neonatal hypoxia-ischemia are unknown, but the expression of *amyloid protein precursor* processing and *tau protein* genes, major factors in Alzheimer’s disease, have been shown to increase in an animal model of an ischemic brain [24,25,26,27,28,29]. In addition, another study showed a significant decrease in the amount of β-amyloid peptide 1–42 in the cerebrospinal fluid post-asphyxia in newborns [31]. Based on the amyloid hypothesis of Alzheimer’s disease, the reduction of the β-amyloid peptide 1–42 in the cerebrospinal fluid is believed to be the first significant change in this disease [32]. In the prodromal and preclinical stages of Alzheimer’s disease, levels of β-amyloid peptide 1–42 in the cerebrospinal fluid are decreased [33] a similar pattern of amyloid changes has been reported in experimental perinatal asphyxia [31].

Scientists are currently focusing on developing new tools for the early identification of the severity of perinatal asphyxia, in particular the prediction of short- and long-term sequel, to reduce mortality, neurological deficits and multi-organ damage [3,34]. Despite many efforts to study various possible indicators of the severity of perinatal asphyxia or hypoxic-ischemic encephalopathy by altering inflammatory and neurons-specific proteins, microRNAs, metabolite pathways and others, no biomarkers have been confirmed for severity and especially late outcome so far [35,36]. In this situation, consideration should be given to proteins identified in lymphocytes that are associated with neurodegeneration like Alzheimer’s’ disease, including in particular proteins encoded by the *amyloid protein precursor* and the *presenilin 1* and *2* genes connected with Alzheimer’s disease [37].

Thus, currently, hypoxic-ischemic encephalopathy remains a global health problem. Experts have emphasized the need to improve complementary therapies in hypothermia and to identify and validate biomarkers of both damage and outcome to achieve the best protocol to protect the infant’s brain and reduce disability and death associated with such a condition [38]. The main challenge in the clinic now is the unpredictability of this phenomenon and the fact that after its beginning, little can be done to limit its devastating consequences. Therefore, it is understandable why many preclinical and clinical studies focus on finding early predictors of perinatal asphyxia, as well as determining determinants of early and late asphyxia consequences, on the basis of which postpartum prevention or treatment can be planned. The aim of this article is the influence of perinatal asphyxia on brain damage observed through the preservation of neurodegenerative genes in lymphocytes in the context of the immature central nervous system, and to predict the consequences of progressive pathological changes in short- and long-term survival and to propose further research on this topic. Based on the above, we decided to conduct genomic studies, supplementing the existing metabolomics data, based on the use of RT-PCR to study the consequences and response of lymphocytes in the circulation after perinatal asphyxia. Lymphocytes were analyzed in patients with birth asphyxia to obtain genomic fingerprints or to test selected sets of genes related to neurodegeneration, such as the *amyloid protein precursor*, *β-secretase*, *presenilin 1* and 2 and *hypoxia-inducible factor 1-α* to determine the severity of the disease or to identify the underlying factors for brain neurodegeneration and get information about the time they occurred.

## 2. Results

### 2.1. Expression of the Amyloid Protein Precursor Gene in Lymphocytes

After perinatal asphyxia, the expression of the *amyloid protein precursor* gene in the 1–7, 8–14 and 15+ age groups was below the control values. In the 1–7 day group, the minimum was −1.721-fold change and the maximum −0.104-fold change with a median of −1.063-fold change. In the 8–14 day group, the minimum was −0.886-fold change and the maximum −0.038-fold change with a median −0.484-fold change. In the group over 15 days, the minimum was −0.907-fold change and the maximum −0.031-fold change with a median −0.364 fold change. Figure 1A illustrates the changes in the mean expression level of the *amyloid protein precursor* gene in the lymphocytes of the studied groups. The changes between the groups were not statistically significant (Figure 1A).

### 2.2. Expression of the β-Secretase Gene in Lymphocytes

After perinatal asphyxia, the expression of the *β-secretase* gene in the 1–7, 8–14 and 15+ age groups was below the control values. In the 1–7 day group, the minimum was −1.187-fold change and the maximum −0.139-fold change with a median of −0.642-fold change. In the 8–14 day group, the minimum was −0.693-fold change and the maximum −0.030-fold change with a median −0.440-fold change. In the group over 15 days, the minimum was −1.009-fold change and the maximum −0.033-fold change with a median −0.470-fold change. Figure 1B illustrates the changes in the mean expression level of the *β-secretase* gene in the lymphocytes of the studied groups. The changes between the groups were not statistically significant (Figure 1B).

### 2.3. Expression of the Hypoxia-Inducible Factor 1-α Gene in Lymphocytes

After perinatal asphyxia, the expression of the *hypoxia-inducible factor 1-α* gene in the 1–7, 8–14 and 15+ age groups was below the control values. In the 1–7 day group, the minimum was −0.742-fold change and the maximum −0.063-fold change with a median of −0.388-fold change. In the 8–14 day group, the minimum was −1.398-fold change and the maximum −0.134-fold change with a median −0.271-fold change. In the group over 15 days, the minimum was −0.783-fold change and the maximum −0.040-fold change with a median −0.149-fold change. Figure 1C illustrates the changes in the mean expression level of the *hypoxia-inducible factor 1-α* gene in the lymphocytes of the studied groups. The changes between the groups were not statistically significant (Figure 1C).

### 2.4. Expression of the Presenilin 1 Gene in Lymphocytes

After perinatal asphyxia, the expression of the *presenilin 1* gene in the 1–7 and 8–14 age groups was below the control values. In contrast, in the age group 15+ it was above the control values. In the 1–7 day group, the minimum was −0.731-fold change and the maximum −0.092-fold change with a median of −0.467-fold change. In the 8–14 day group, the minimum was −0.674-fold change and the maximum −0.027-fold change with a median −0.298-fold change. In the group over 15 days, the minimum was 0.145-fold change and the maximum 0.993-fold change with a median 0.305-fold change. Figure 2A illustrates the changes in the mean expression level of the *presenilin 1* gene in the lymphocytes of the studied groups. The changes were statistically significant between the 1–7 and 15+ day groups and between the 8–14 and 15+ day groups after perinatal asphyxia (Figure 2A).

### 2.5. Expression of the Presenilin 2 Gene in Lymphocytes

After perinatal asphyxia, the expression of the *presenilin 2* gene in the 1–7 and 8–14 age groups was below the control values. In contrast, in the age group 15+ it was above the control values. In the 1–7 day group, the minimum was −1.495-fold change and the maximum −0.263-fold change with a median of −0.529-fold change. In the 8–14 day group, the minimum was −1.328-fold change and the maximum −0.349-fold change with a median −0.625-fold change. In the group over 15 days, the minimum was 0.107-fold change and the maximum 1.068-fold change with a median 0.396-fold change. Figure 2B illustrates the changes in the mean expression level of the *presenilin 2* gene in the lymphocytes of the studied groups. The changes were statistically significant between the 1–7 and 15+ day groups and between the 8–14 and 15+ day groups after perinatal asphyxia (Figure 2B).

## 3. Discussion

Here we show, we believe for the first time, that circulating lymphocytes in post-asphyxia neonates overexpress the *presenilin 1* and *2* genes with age, which genes are associated with neurodegenerative diseases i.e., post-ischemic brain injury and Alzheimer’s disease. Moreover, we found that the expression of the *amyloid protein precursor, β-secretase* and *hypoxia-inducible factor 1-α* gene was below the control values.

It was documented that activated lymphocytes can routinely penetrate the central nervous system, in cases of brain ischemia and Alzheimer’s disease and participate in neuroinflammatory process [37,39,40,41,42]. Increased expression of *presenilin* genes related to γ-secretase metabolizing amyloid protein precursor to amyloid positively correlates with the staining presenilin protein and amyloid in lymphocytes in neurodegenerative diseases [37]. Analysis of immunopeptide databases provides evidence that neurodegeneration associated proteins present in human lymphocytes are a source of endogenous neurodegenerative proteins in brain like amyloid protein precursor, presenilin 1 and 2, tau protein, apolipoprotein E and α-synuclein [37]. Lymphocytes responsiveness to β-amyloid peptide staining product of γ-secretase was reported first time over twenty years ago in Alzheimer’s disease patients’ brains [37,39,40].

A key player in cellular response to hypoxia-ischemia is hypoxia-inducible factor 1-α, a transcriptional regulator of numerous genes involved in adaptive and survival mechanisms of neurons transiting from normoxic to hypoxic conditions [37]. However, in the case of prolonged or severe ischemia, hypoxia-inducible factor 1-α may exert deleterious effects such as apoptosis and inhibition of cell growth [37], which positively correlates with our observations of decreased expression in post-asphyxia lymphocytes, the *hypoxia-inducible factor 1-α* gene.

In subjects with vascular dementia, healthy elderly persons and Alzheimer’s disease patients, there were no significant changes in the lymphocytic amyloid protein precursor and β-secretase mRNA, but their mRNA were present in lymphocytes [43,44]. Above data positively correlates with our observations of decreased expression in post-asphyxia lymphocytes, the *amyloid protein precursor* and *β-secretase* genes.

We have shown that the levels of overexpression of the γ-secretase-related *presenilin* genes in lymphocytes begin to increase 15 days after the asphyxia episode, suggesting the possible ability of the presenilins to reflect pathological processes in the injured brain that express the systemic nature of the disorder. Based on this preliminary study of presenilin expression as a possible marker of brain neurodegeneration, the activation of these genes as well as their potential as a marker of brain neurodegeneration should be considered. The understanding and neurological prognosis of hypoxic-ischemic encephalopathy post-asphyxia is limited. There is a need for early and accurate prognostic methods, both to avoid long-term treatment of patients who fail further treatment and to ensure optimal treatment for patients with a potential for recovery. Taken together, our results indicate that increased expression of *presenilin* genes in lymphocytes may be a promising biomarker for predicting/prognosticating the severity of post-asphyxia brain neurodegeneration and potentially useful for predicting poor sequel of asphyxia later in life.

## 4. Materials and Methods

### 4.1. Study Setting and Design

Our main goal was to use knowledge to predict brain damage and monitor treatment outcomes after perinatal asphyxia, based on the assumption that the pathogenesis and regeneration of neurons can be interpreted in terms of modulating effects from outside the brain, i.e., lymphocytes. Therefore, we conducted clinical trials of both newborns with the history of perinatal asphyxia and healthy ones which constituted the control group, born in the Lublin Province, Poland. At the time of inclusion in the study, all patients were hospitalized in the Department of Neonate and Infant Pathology of Medical University of Lublin. The inclusion criteria for the asphyxia group were defined as follows (inclusion criteria refer to symptoms observed in the first day of life):Newborns (full-term and preterm) > 31 weeks of gestational age,metabolic acidosis with pH < 7.0 (in umbilical cord or newborn blood sample obtained within 60 min after birth),or Base deficit (BE) > −12,or Apgar score of 0–5 at 10 min or continued need for resuscitation at 10 min of age,and presence of multiple organ-system failures,and clinical evidence of encephalopathy: hypotonia, abnormal oculomotor or pupillary movements, weak or absent suck, periodic breathing/apnea or clinical seizures,neurologic findings cannot be attributed to other cause (inborn error of metabolism, a genetic disorder, congenital neurologic disorder, medication effect).

Each time after obtaining the consent of the patient’s legal guardian in both groups, blood was collected along with blood sampling for routine laboratory tests necessary for diagnosis or treatment. The study protocol was approved by the Ethics Committee of the Medical University of Lublin (25 April 2013, Consent no. KE−0254/118/2013). Written informed consent was obtained from legal guardians of all participants.

### 4.2. Study Population and Sample Size

The study included 52 newborns, 26 newborns after perinatal asphyxia (the study group) and 26 healthy newborns born without perinatal complications (control group). The test and control groups were divided into three age groups 1–7 days, 8–14 days and 15+ days. In the study group, aged 1–7 days, there were 8 newborns, including 6 girls and 2 boys. In the age group of 8–14 days there were 7 newborns, including 5 girls and 2 boys, and in the group over 15 days there were 11 children, including 6 girls and 5 boys.

In the control group, aged 1–7 days, there were 8 children, including 2 girls and 6 boys. All the children were hospitalized due to hyperbilirubinemia, with the final diagnosis of breast milk jaundice. In the age group of 8–14 days there were 7 children, including 3 girls and 4 boys. The children were hospitalized for breast milk jaundice, respiratory infections, allergic rash and infantile colic. In the group of 15+ 11 infants, including 7 girls and 4 boys. Children were hospitalized for breast milk jaundice, infantile colic, respiratory infection and poor weight gain. Children in the control group were born after uncomplicated pregnancy and labor and without any serious health problems. Children of these groups were hospitalized in the Department of Neonate and Infant Pathology for general evaluation or diagnosis and treatment due to symptoms described above. Clinical characteristics of enrolled infants in study are presented in Table 1 and Figure 3.

### 4.3. Study of Gene Expression Such as: Amyloid Protein Precursor, β-Secretase, Presenilin 1 and 2 and Hypoxia-Inducible Factor 1-α

For lymphocyte studies, venous blood was collected for citrate. After collection, blood was placed in a centrifuge tube (Falcon) and diluted 1:1 with PBS (Biomed, Poland). The blood prepared in this way was layered on the Gradisol L reagent (Polfa, Poland). Then the sample was centrifuged in a Centrifuge 5810R (Eppendorf, Germany) at 2000 rpm for 20 min at room temperature. Peripheral blood mononuclear cells were harvested and washed in 5 mL PBS and then centrifuged at 2000 rpm for 10 min at room temperature. After that, the cells were suspended in 1 mL PBS and centrifuge again in a MiniSpin plus (Eppendorf) centrifuge at room temperature at 2000 rpm for 10 min. Subsequently, RNA was isolated from the pellet. RNA isolation was performed according to the method of Chomczyński and Sacchi [45,46]. After pouring into the sediment of cold TRI (4 °C) (Invitrogen) with a volume of 0.5 mL, the material was homogenized manually. Then 0.2 mL of chloroform was added to each sample, shaken for 15 s and left at room temperature for 15 min. The samples were then centrifuged in the Centrifuge 5415 R at 4 °C at 13,600 rpm for 15 min and total cellular RNA was precipitated from the aqueous phase by adding 0.5 mL of isopropyl alcohol. Finally, the samples were centrifuged in the Centrifuge 5415 R at 4 °C at the speed of 13,600 rpm for 20 min. The Nano Drop 2000 spectrophotometer (Thermo Fisher Scientific, Waltham, MA, USA) was used to assess the quality and quantity of RNA. The obtained RNA was stored in an 80% ethanol at −20 °C.

In further studies, 1 μg of total RNA was reverse transcribed into cDNA using a high-capacity cDNA kit for reverse transcription according to the manufacturer’s instructions (Applied Biosystems, USA). The cDNA synthesis was performed on Veriti D × (Applied Biosystems, Foster City, CA, USA) under the following conditions: stage I 25 °C, 10 min; stage II 37 °C, 120 min; stage III 85 °C, 5 min; stage IV 4 °C. The cDNA, obtained by this procedure, was amplified by real-time gene expression analysis (qPCR) on a 7900HT Real-Time Fast System (Applied Biosystems, Foster City, CA, USA) with the Master Mix SYBRgreen PCR power mix reagent, using the manufacturer’s SDS software [46]. The amplification protocol included the following cycles: initial denaturation 95 °C, 10 min and 40 cycles, each at two different temperatures 95 °C, 15 s and 60 °C, 1 min. The monitoring and calculation of the number of DNA copies was carried out in the 7900HT Real-Time Fast System (Applied Biosystems, Foster City, CA, USA) in each amplification cycle. The number of cycles of PCR at which the fluorescence level exceeded the specific relative expression of the threshold cycle (CT) was applied to the research software (Applied Biosystems, Foster City, CA, USA) to calculate the number of DNA molecules present in the mixture at the beginning of the reaction. Normalization was achieved against the endogenous control gene glyceraldehyde 3-phosphate dehydrogenase (GAPDH), and the relative quantification (RQ) of the gene expression was analyzed based on the ΔCT method and the results were calculated as RQ = 2^−ΔΔCT^ [46,47]. The RQ values were finally analyzed after their logarithmic conversion to the RQ logarithm (LogRQ) [46]. LogRQ = 0 indicates that gene expression in control and ischemic samples is not different. LogRQ < 0 means that we have reduced gene expression in the asphyxia sample, while LogRQ > 0 indicates increased gene expression in the asphyxia sample compared with the control one. The following sets of TaqMan (Applied Biosystems, Foster City, CA, USA) probes labeled with FAM-NFQ: Hs00997789_m1 (*PSEN1*), Hs01577197_m1 (*PSEN2*), Hs99999905_m1 (*GAPDH*) and Hs00169098_m1 (*APP*), Hs01121195_m1 (*BACE1*) and Hs00153153_m1 (*HIF1α*) were used. Statistical evaluation of the results was carried out using the Statistica v. 13.3 software (Tibco Corporation, Palo Alto, CA, USA) with the help of non-parametric Kruskal-Wallis test with the “z” test-multiple analyses of differences between groups. Data are presented as mean ± SD. Statistical significance was adopted at *p* ≤ 0.05.

## 5. Conclusions

Lymphocytes are best known as primary mediators of inflammation generation; however, new emerging evidence demonstrates that lymphocytes are far more complex than previously regarded, equipped with genetically complex intracellular machinery. Now recognized as key players in inflammation and immune responses, these cells express and secrete several pro-neurodegenerative molecules that serve to initiate and modulate neurodegeneration [37]. Understanding lymphocyte signals in states of neurodegeneration and how they modulate neurodegeneration could extend new diagnostic and therapeutic approaches to monitoring and combating diseases associated with progressive neurodegeneration. It is necessary to detect factors responsible for brain neurodegeneration and obtain information about their timing. Delayed diagnosis of neurodegeneration is a major obstacle to finding an effective treatment. In this sense, the development of new molecular biomarkers is essential for diagnostic and prognostic purposes. Our genomic study suggested that the *presenilin* genes have enormous prognostic and diagnostic potential and can shed new light on the pathogenesis pathways in many neurodegenerative diseases with amyloid accumulation.

Since the main and ultimate goal of imaging brain pathology is to use knowledge to predict disease and monitor treatment outcomes, it is likely that neuronal pathogenesis and regeneration can be interpreted in terms of modulator effects coming from outside the brain in this situation from lymphocytes. Research into biomarkers of neonatal brain neurodegeneration is currently at a very early stage of development, but there is still no significant progress to be made. Presenilins are γ-secretase related proteins, and γ-secretase is required for the production of amyloid from the amyloid protein precursor. *Presenilin* genes overexpression, are present in both neurons and neuroglial cells. Overexpression of the *presenilin* genes in lymphocytes may indicate indirect neuronal damage in diseases of the central nervous system such as hypoxic-ischemic encephalopathy or ischemic brain neurodegeneration. Examination by MRI showed that lymphocyte infiltrates were present at a late stage after experimental ischemic brain injury for more than 1 year [48]. This was confirmed by immunostaining of lymphocytes present in the post-ischemic hippocampus and striatum [48]. In addition, infiltration of lymphocytes into the brains of patients was observed in postmortem studies in the late stages of ischemic stroke [41]. These observations demonstrated persistent dysfunction of the blood-brain barrier in the late stages after cerebral ischemia [48], which in the long term may lead to prolonged infiltration of lymphocytes with overexpressed *presenilin* genes, which are a component of the γ-secretase used for amyloid production. Activation and expression of effectors molecules in these cells in the late period after asphyxia suggests their potential involvement in the late stages of brain hypoxia-ischemia encephalopathy [41]. These findings could provide a better knowledge base for future advanced research to unveil the precise role of lymphocytes responses in the late stage of hypoxia-ischemia [41]. Thus, the balance of degenerative processes and the control of inflammation by lymphocytes with neurogenesis may determine the long-term outcomes of global survival due to hypoxia-ischemia or amyloid plaques formation with Alzheimer’s disease-type dementia.

The possibility of measuring the whole metabolomics and genomics profile and the screening of molecular intermediates of different pathological pathways give a multiparametric response that appears to be more adequate for facing any complex and multifactor disease. Moreover, the study of the dynamic changes of genomics profiles can help to improve different crucial clinical aspects, including early perinatal asphyxia diagnosis and full-blown development of hypoxic-ischemic encephalopathy, response to treatments and, possibly, the prediction of early and late outcomes. The current need is to define a profile or a set of biomarkers that could be used in a narrower temporal window to help clinicians in use early neuroprotection. Prospective long-term follow-up studies are needed to verify lymphocytes gene changes’ predictive role. This finding creates the need for further research to better understand the role of *presenilin* genes changes expression in lymphocytes with age in the late phase of post-asphyxia. Whether this constitutes an etiological basis for neurodegenerative disorder such as Alzheimer’s disease in adulthood should be explained through additional experiments as well as clinical and epidemiological observations [24]. Therefore, while the detailed relationships between neuroimaging results and post-ischemic histological changes remain unclear, understanding these relationships or developing methods to visualize histopathological changes in one cell using human brain neuroimaging would be helpful in considering future indications for therapy during reperfusion in cases of hypoxic-ischemic encephalopathy. This appears to be the first worldwide data on the role of the *presenilin 1* and *2* genes associated with Alzheimer’s disease in the dysregulation of neonatal lymphocytes after perinatal asphyxia.

## Figures and Tables

**Figure 1 ijms-22-05140-f001:**
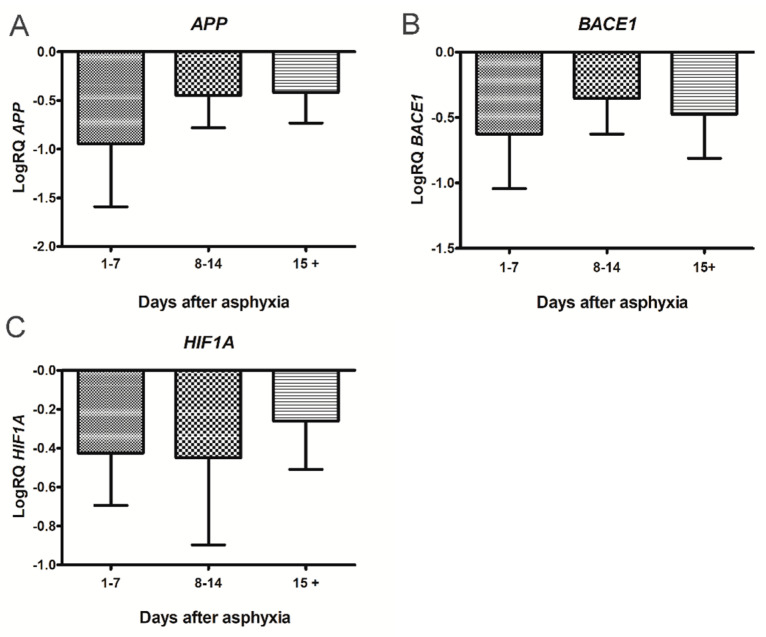
(**A**) Mean levels of *amyloid protein precursor* (*APP*) gene expression in lymphocytes after perinatal asphyxia in age groups 1–7 (*n* = 8), 8–14 (*n* = 7) and 15+ (n = 8) days. Marked SD, standard deviation. N, number of children in the group. The changes between the groups were not statistically significant (Kruskal-Wallis test). (**B**) Mean levels of *β-secretase* (*BACE1*) gene expression in lymphocytes after perinatal asphyxia in age groups 1–7 (*n* = 8), 8–14 (*n* = 7) and 15+ (*n* = 8) days. Marked SD, standard deviation. N, number of children in the group. The changes between the groups were not statistically significant (Kruskal-Wallis test). (**C**)**.** Mean levels of *hypoxia-inducible factor 1-α* (*HIF1A*) gene expression in lymphocytes after perinatal asphyxia in age groups 1–7 (*n* = 8), 8–14 (*n* = 7) and 15+ (*n* = 9) days. Marked SD, standard deviation. N, number of children in the group. The changes between the groups were not statistically significant (Kruskal-Wallis test).

**Figure 2 ijms-22-05140-f002:**
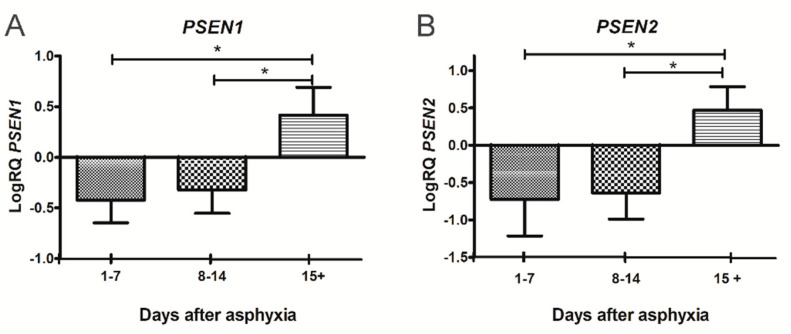
(**A**). Mean levels of *presenilin 1* (*PSEN1*) gene expression in lymphocytes after perinatal asphyxia in age groups 1–7 (*n* = 8), 8–14 (*n* = 7) and 15+ (*n* = 10) days. Marked SD, standard deviation. N, number of children in the group. The changes between the 1–7 and 8–14 day groups were not statistically significant. The indicated statistically significant differences in the level of gene expression between groups 1–7 and 15+ days (*z* = 3.903, *p* = 0.0002) and between 8–14 and 15+ days (*z* = 3.092, *p* = 0.0059) after perinatal asphyxia (Kruskal-Wallis Test). * *p* ≤ 0.01. (**B**). Mean levels of *presenilin 2* (*PSEN2*) gene expression in lymphocytes after perinatal asphyxia in age groups 1–7 (*n* = 8), 8–14 (*n* = 7) and 15+ (*n* = 10) days. Marked SD, standard deviation. N, number of children in the group. The changes between the 1–7 and 8–14 day groups were not statistically significant. The indicated statistically significant differences in the level of gene expression between groups 1–7 and 15+ days (*z* = 3.634, *p* = 0.0008) and between 8–14 and 15+ days (*z* = 3.387, *p* = 0.002) after perinatal asphyxia (Kruskal-Wallis Test). * *p* ≤ 0.01.

**Figure 3 ijms-22-05140-f003:**
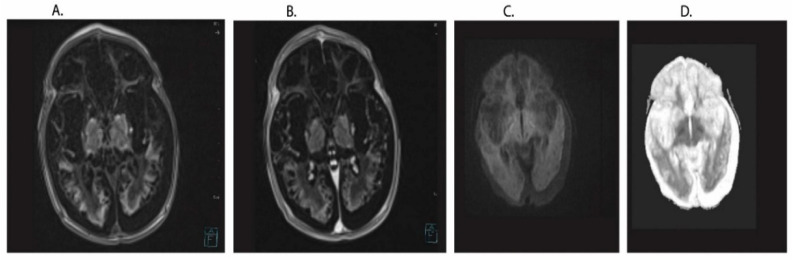
Sample MRI images one of the studied patient from group 15+ T1 (**A**,**B**), DWI (**C**) and ADC (**D**) after neonatal asphyxia. Generalized atrophy of the cortex with widening of the fluid space mainly in the occipital area. Features of generalized encephalomalacia with the presence of small fluid spaces. The picture of the brain structures corresponds to hypoxic-ischemic changes.

**Table 1 ijms-22-05140-t001:** Clinical characteristics of enrolled infants.

Group	Age (Days)	Gestational Age (Weeks)	Birth Weight (g)	Apgar Score (1 min)	RBC (×1000/µL)	WBC (/µL)	Lymphocyte (/µL)	PLT (×1000/µL)	Hct (%)	pH	BE (Mmol/L)
**1–7 days after birth *n* = 8/group**											
Control	6 ± 1	39 ± 1	3082 ± 369	10.0 ± 0	4845 ± 184	12,661 ± 1133	6713 ± 860	405 ± 30	45 ± 1	7.40 ± 0.03	1.0 ± 0.8
Asphyxia	4 ± 3	39 ± 2	3721 ± 501	3.4 ± 2	4651 ± 663	25,988 ± 8632	4541 ± 2507	259 ± 66	50 ± 8	7.25 ± 0.12	−5.8 ± 5.5
**8–14 days after birth *n* = 7/group**											
Control	12 ± 2	39 ± 1	3539 ± 723	9.9 ± 0.4	4964 ± 366	12,317 ±3851	7007 ± 1070	323 ± 102	47 ± 4	7.39 ± 0.03	1.4 ± 2.4
Asphyxia	12 ± 2	39 ± 3	3153 ± 862	2.7 ± 2.5	4470 ± 462	20,181 ± 4016	6311 ± 1671	236 ± 35	46 ± 5	6.99 ± 0.23	−12.2 ± 9.8
**15+ days after birth *n* = 11/group**											
Control	20 ± 3	38 ± 2	3343 ± 558	9.8 ± 0.4	4534 ± 489	13,254 ± 2021	7129 ± 1310	601 ± 126	41 ± 4	7.41 ± 0.02	0.2 ± 1.3
Asphyxia	19 ± 3	37 ± 3	2868 ± 622	1.6 ± 1.6	4385 ± 951	18,697 ± 8925	5736 ± 1429	230 ± 54	43 ± 9	7.13 ± 0.25	−13.7 ± 9.1

Data are mean ± SD. RBC—red blood cells, WBC—white blood cells, PLT—platelets, Hct—hematocrit, BE—base deficit.

## Data Availability

The data presented in this study are available on request from the corresponding author.

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
