# Peer review of "Alzheimer’s Disease Associated Presenilin 1 and 2 Genes Dysregulation in Neonatal Lymphocytes Following Perinatal Asphyxia"

_ijms, 2021, doi:10.3390/ijms22105140_

Round 1
Reviewer 1 Report
Review for “Alzheimer's disease associated presenilin 1 and 2 genes dysregulation in neonatal lymphocytes following perinatal asphyxia” by Tarkovska et al. (ijms-1207818).
the present manuscript of describes expression profiles of genes . It is well written and understandable. However, I feel it could be more focused in the introduction and discussion section. The ,amuscript addresses a broad readership and communicates an interesting and important topic in neonatoloy. It has to be pointed out, that acquisition of samples from severely sick newborns is difficult. In this regard the relatively high number of samples increases the validity of the manuscript.
Major suggestions/concerns
- For presenilin-1 1nd-2, an assessment on protein level should be presented to confirm/clarify whether the results reflect mRNA or post-transcriptional regulation. Results on protein levels would support the authors suggestion to use presenilin as a potential diagnostic marker.
- The authors provide LogRQ values to compare control vs. asphyxia groups. It would be interesting to clarify, whether for example APP in the asphyxia groups (8-14/15+) differ significantly from normoxia groups. Do the authors`also check by other test methods (one way ANOVA) or used a post-test (Bonferroni)?
Minor suggestions
1) the introduction should become shorter.
- the figures could be summarized. I suggest two figures for the results which show no significant differences in mRNA expression (APP, BACE1, HIF-1alpha) and such for significant differences in mRNA expression (PRSEN1 and PRESEN2).
Yours sincerly
Author Response
Reviewer 1. Changes are in red and/or highlighted in yellow.
Review for “Alzheimer's disease associated presenilin 1 and 2 genes dysregulation in neonatal lymphocytes following perinatal asphyxia” by Tarkovska et al. (ijms-1207818). The present manuscript of describes expression profiles of genes . It is well written and understandable. However, I feel it could be more focused in the introduction and discussion section. The manuscript addresses a broad readership and communicates an interesting and important topic in neonatology. It has to be pointed out, that acquisition of samples from severely sick newborns is difficult. In this regard the relatively high number of samples increases the validity of the manuscript. Thank you for recognizing our research efforts. We shortened the introduction by half a page, the discussion by a third of a page, and the conclusions by a quarter of a page.
Major suggestions/concerns
• For presenilin-1 and-2, an assessment on protein level should be presented to confirm/clarify whether the results reflect mRNA or post-transcriptional regulation. Results on protein levels would support the authors suggestion to use presenilin as a potential diagnostic marker.
The proposal is interesting and we will take it into account in the planned future research. At present, we do not have the material available, we have used all the material for gene research. So it requires a renewal of material collection which takes months and sometimes it flies.
• The authors provide LogRQ values to compare control vs. asphyxia groups. It would be interesting to clarify, whether for example APP in the asphyxia groups (8-14/15+) differ significantly from normoxia groups. Do the authors`also check by other test methods (one way ANOVA) or used a post-test (Bonferroni)?
The answer to this remark is quite complicated. The RQ calculations were made in the ExpressionSuite software version 1.1 (2016) program created by Thermo Fisher Scientific. The program is dedicated to devices of this company that perform PCR analyzes. RQ is calculated automatically based on Livak's formula. In the case of the analysis presented in the article, the endogenous control was the GAPDH gene. In order to calculate the RQ, the program compares the expression in the test and control material. The obtained RQ values finally are converted to LogRQ values by means of the decimal log (10) for a better/readable presentation of the results. Thus, comparative analyzes of LogRQ values of the studied genes in 3 groups of newborns of different ages were performed in the Statistica version 13.3 program by TIBCO using the H Kruskal-Wallis test with intergroup comparisons. The performed analysis showed statistically significant differences only in the case of two genes PSEN1 and PSEN2. The significance of the differences for the PSEN1 and PSEN2 genes are lower than the critical level of significance using the Bonferroni correction, i.e. by 0.016. As suggested by the reviewer, we also performed one way ANOVA analysis with Tukey's test. The analysis also showed significant statistical differences in LogRQ only for the PSEN1 and PSEN2 genes. The significance of the differences in the case of the PSEN1 and PSEN2 genes are lower than the critical level of significance using the Bonferroni correction, i.e. by 0.016 . To sum up, in all the used tests, statistical significance is only observed in the case of the PSEN1 and 2 genes.
Minor suggestions
1) The introduction should become shorter.
We shortened the introduction by half a page, the discussion by a third of a page, and the conclusions by a quarter of a page
• The figures could be summarized. I suggest two figures for the results which show no significant differences in mRNA expression (APP, BACE1, HIF-1alpha) and such for significant differences in mRNA expression (PRSEN1 and PRESEN2).
We grouped the figures as suggested by the reviewer. Now we have only 2 gene figures, the first is APP, BACE1 and HIF-1alpha and the second is PSEN1 and PSEN2.
Reviewer 2 Report
Paper present data on the dysregulation of AD associated presenilin 1 and 2 genes in neonatal lymphocytes suffered from perinatal asphyxia: This is a novel approach which epidemiologically can be used not only for predicting/prognosticating the severity of post-asphyxia brain neurodegeneration or for predicting poor sequelae of asphyxia later in life, but also potentially in the adult/aged patient AD ethiology. Design of the study is well structured, methodology can bring novel results and experimental outcomes seems sound. Discussion is adequeate, however other risk metabolic factors such as homocysteine can also be noted in both the mothers as well as within the childhood or adult life periods.
Author Response
Reviewer 2 Changes are in red and/or highlighted in yellow. Paper present data on the dysregulation of AD associated presenilin 1 and 2 genes in neonatal lymphocytes suffered from perinatal asphyxia: This is a novel approach which epidemiologically can be used not only for predicting/prognosticating the severity of post-asphyxia brain neurodegeneration or for predicting poor sequelae of asphyxia later in life, but also potentially in the adult/aged patient AD etiology. Design of the study is well structured, methodology can bring novel results and experimental outcomes seems sound. Discussion is adequate, however other risk metabolic factors such as homocysteine can also be noted in both the mothers as well as within the childhood or adult life periods. Thank you for your positive evaluation of our research efforts. In future research, which we plan to continue, we will consider the reviewer's suggestions regarding the assessment of homocysteine in children and mothers.
Round 2
Reviewer 1 Report
Dear Editor(s),
I have read the answer letter of the authors and I feel that all points were addressed carefully. I understand that it is not applicable to collect fresh samples to provide protein analysis.. This is regrettable, but the results are important and interesting as they stand for it now.
Therefore, I would like to recommend the present version of the manuscript for publication in your journal.
Best regards